# Bioacaricidal Potential of *Moringa oleifera* Ethanol Extract for *Tetranychus merganser* Boudreaux (Acari: Tetranychidae) Control

**DOI:** 10.3390/plants10061034

**Published:** 2021-05-21

**Authors:** Rapucel Tonantzin Quetzalli Heinz-Castro, Roberto Arredondo-Valdés, Salvador Ordaz-Silva, Heriberto Méndez-Cortés, Agustín Hernández-Juárez, Julio Cesar Chacón-Hernández

**Affiliations:** 1Faculty of Agronomy and Veterinary, Universidad Autónoma de San Luis Potosí, Soledad de Graciano Sánchez 78321, San Luis Potosí, Mexico; rapucel.heinz@uaslp.mx (R.T.Q.H.-C.); heriberto.mendez@uaslp.mx (H.M.-C.); 2Faculty of Chemical Science, Universidad Autónoma de Coahuila, Saltillo 25280, Coahuila, Mexico; r-arredondo@uadec.edu.mx; 3Faculty of Business and Engineering San Quintín, Universidad Autónoma de Baja California, San Quintín 22930, Baja California, Mexico; salvador.ordaz.silva@uabc.edu.mx; 4Parasitology Department, Universidad Autónoma Agraria Antonio Narro, Saltillo 25315, Coahuila, Mexico; chinoahj14@hotmail.com; 5Institute of Applied Ecology, Universidad Autónoma de Tamaulipas, Victoria City 87019, Tamaulipas, Mexico

**Keywords:** red spider mite, oviposition, hatched eggs, damage, feed intake, mortality, residual

## Abstract

The Tetranychidae family includes mites causing severe damage to agricultural fields. The red spider mite, *Tetranychus merganser* Boudreaux (Acari: Tetranychidae), causes severe damage to several plant species grown as cash crops. Current red spider mite control depends mainly on chemical insecticides. There is a need for alternate control measures that are environmentally friendlier than chemical pesticides. The aim of the study was to assess the effects of *Moringa oleifera* leaf ethanolic extract at different concentrations (0.1, 0.5, 1, 5, 10, 15, and 20% (*v*/*v*)) against *T. merganser* females. Such effects can serve as a basis to include this compound in integrated pest management programs for the control of red spider mites. Mites treated with 20% (*v*/*v*) killed 86.67%, 13.70%, and 96.30% at 24, 48, and 72 h, respectively, as compared to the control treatment. Oviposition, egg hatching, and the damage caused by red spider mites were all reduced at high concentrations. *Moringa oleifera* leaf ethanolic extract can be used as a powerful bioacaricide for the control of *T. merganser*.

## 1. Introduction

Tetranychidae are distributed all over the world, encompassing a little more than 1300 mite species, clustered in over 70 genera that are divided into two sub-families, including Bryobinae and Tetranychinae [1,2,3]. Most agricultural mite pests belong to the Tetranychinae sub-family [1,2]. Species within the *Tetranychus* Dufour genus tend to group in colonies wrapped around in silk. Some of these species show extreme polyphagia and they thrive optimally in a large number of plant species, leading to a widespread global distribution [4]. The Red Spider Mite, *Tetranychus merganser* Boudreaux (Acari: Tetranychidae), is distributed in China, USA, Australia, and Mexico, where it attacks different families of plant hosts, including Solanaceae, Rosaceae, Oleaceae, Acuifoliaceae, Ranunculaceae, Caricaceae, and Cactaceae. This mite has a high reproductive potential and short life cycle [5,6]. The damage that *T. merganser* causes to plants can begin with the development of white spots close to the foliar veins. After some time, and under severe infestations, these spots can merge, turning the leaves white [7].

Arthropod pest control with chemical insecticides faces economic and ecological challenges all over the world, due to the impact on human health and on the environment. The identification of new alternatives is crucial to fight the resistance that arthropod pests have developed to chemical pesticides. Plant extracts are environmentally and human-friendly options that can help manage agricultural pests [8,9,10].

The tree *Moringa oleifera* Lamarck (Moringaceae) has agricultural importance in Mexico and in other tropical countries of the world [11]. People eat the fruits, seeds, and roots. The stalks are used as animal feed and the seed oils are part of lubricants, cosmetics, and biofuels [12,13]. The original research works that studied the different parts of the *M. oleifera* tree show different medicinal properties, such as anti-tumor, anti-pyretic, anti-epileptic, anti-inflammatory, anti-ulcer, anti-diabetic, anti-oxidizing, anti-fungal, anti-bacterial, and anti-carcinogenic effects [14]. Several research works have reported the insect pest control properties of *M. oleifera*. Furthermore, *M. oleifera* oils have shown to produce an anti-feeding effect on *Spodoptera frugiperda* Walker larvae (Lepidoptera: Noctuidae) [15] and *Spodoptera littoralis* Boisd (Lepidoptera: Noctuidae) [16]. The leaf powder of *M. oleifera* produces anti-egg-laying activity on *Callosobruchus maculatus* F. (Coleoptera: Chrysomelidae) [17] and *Tribolium castaneum* Herbst (Coleoptera: Tenebrionidae) [18]. By the same token, lectins in *M. oleifera* seeds have larvicidal properties against the Mediterranean flour moth *Ephestia kuehniella* Zeller (Lepidoptera: Pyralidae) [19] and mosquito *Aedes aegypti* L. (Diptera: Culicidae) [20]. The aim of this research work was to assess the effects of *M. oleifera* leaf ethanolic extract at different concentrations on *T. merganser*, in order to include this compound in integrated pest management (IPM) programs for the control of red spider mites.

## 2. Results

### 2.1. Phytochemical Analysis

The phytochemical analysis of the *M. oleifera* leaf ethanolic extract showed several groups of secondary metabolites, such as phenols, alkaloids, flavonoids, tannins, saponins, carbohydrates, and quinones, among others (Table 1).

### 2.2. Acaricidal Effect of M. oleifera Extract on T. merganser

#### 2.2.1. Mortality

The ethanolic extract of *M. oleifera* leaf showed acaricidal activity at different concentrations and different times against *T. merganser* females. The concentrations (0.1, 0.5, 1, 5, 10, and 20% (*v*/*v*)) caused acute toxicity to female mites at 24 h (F = 47.50; df = 6, 14; *p* < 0.0001), 48 h (F = 63.58; df = 6, 14; *p* < 0.0001), and 72 h (F = 94.05; df = 6, 14; *p* < 0.0001). Red spider mites treated with 0.1 and 20% (*v*/*v*) killed 0.00% and 46.67%, 6.67% and 86.67%, and 13.70% and 96.30%, at 24, 48, and 72 h, respectively, as compared to the control group (Table 2), showing that *T. merganser* mortality depends on the applied concentration rates. 

#### 2.2.2. Toxicity of the Extract 

Table 3 shows the relative toxicity results of the *M. oleifera* leaf ethanolic extract. The values of LC_50_ and LC_90_ were 6.06% and 17.59% (*v*/*v*), indicating that 50% and 90% of *T. merganser* population died at these concentration rates, respectively.

#### 2.2.3. Oviposition

Egg production by *T. merganser* females that could still oviposit was significantly different at 24 (F = 940.69; df = 7, 16; *p* < 0.0001), 48 (F = 2070.21; df = 7, 16; *p* < 0.0001), and 72 h (F = 4026.22; df = 7, 16; *p* < 0.0001). Females treated with 0.1 and 20% (*v*/*v*) laid a lower number of eggs: 2.54 to 94.69%, 17.35 to 96.31%, and 23.18 to 97.63% less than the control group at 24, 48 and 72 h, respectively. The growth rate of the spider mite was significantly different among the treatments (F = 681.08; df = 7, 16; *p* < 0.0001) and it decreased as the concentration increased (Table 4).

#### 2.2.4. Hatched Eggs

The number of eggs hatched on the fourth day was significantly different among the treatments (F = 1227.42; df = 7, 16; *p* < 0.0001). The percentage of hatched eggs in the control group was 96.74%; while in treated females, this percentage ranged from 79.46% (0.1% (*v*/*v*)) to 0.00% (20.0 % (*v*/*v*)) (Table 5). This is an indication that the different concentrations had a residual effect on the eggs laid during the first 24 h, which were supposed to hatch four days after being laid.

#### 2.2.5. Food Intake

The percentage of damage was significantly different among treatments at 24 (F = 58.04; df = 7, 16; *p* < 0.0001), 48 (F = 367.99; df = 7, 16; *p* < 0.0001), and 72 h (F = 298.32; df = 7, 16; *p* < 0.0001). *Tetranychus merganser* females treated with 0.1 and 20% (*v*/*v*) decreased their feed intake by 13.05 and 80.31%, 13.27 and 84.19%, and 7.94 and 85.48% at 24, 48, and 72 h, respectively, in comparison to the control group (Table 6).

## 3. Discussion

Just like in other reports, in this work we found that *M. oleifera* leaf ethanolic extract has different bioactive ingredients, including flavonoids, phenols, alkaloids, carotenoids, tannins, saponins, and carbohydrates. This result coincides with several past research works [14,21,22]. Some phytochemicals in the plant, such as flavonoids, alkaloids, glycosides, esters, and fatty acids, affect the behavior (repelling, deterring/anti-feeding, and alluring) as well as the physiology (toxic, growth retarding, and chemo sterilizing) of several insect and herbivore species [23]. These effects are the result of several toxic compounds acting in synergy within the extract [24].

The literature does not refer to other studies on the use of *M. oleifera* for the control of *Tetranychus* spp. However, other studies have researched the anti-insect properties of different *M. oleifera* leaf and seed extracts, and their findings are very similar to ours. This research work showed that the mortality of *T. merganser* increases as the concentration of *M. oleifera* extract increases. Likewise, the egg-hatching, growth rate, oviposition, and feeding rates decrease with higher concentrations of the ethanolic extract. Moawad and Sadek [16] reported that a 10% (*v*/*v*) concentration rate of *M. oleifera* oil kills 66 and 76% of the first and third larval instars of *Spodoptera littoralis* in 24 h. Kamel [15] found that an increase in the mortality (42.2–100%) of *Spodoptera frugiperda* was due to the concentration rates (1.25–10% (*v*/*v*)) of *M. oleifera* oil. On the other hand, Adenekan et al. [25] reported that pulverized leaves and stems of *M. oleifera* (0.5 g of powder with 30 g of cowpea seeds, *Vigna unguiculata* Walp (Fabaceae)) caused 85% and 70% mortality in *Callsobruchus maculatus* after 24 h. On the other hand, Ojo et al. [17] reported that 2 g of *M. oleifera* leaf powder with 20 g of cowpea seeds led to 36.33% mortality in *Callosobruchus maculatus*. Anita et al. [18] found that a concentration of 2.0 g of *M. oleifera* per 10 g of wheat (*Triticum aestivum* L. (Poaceae)) kills 100% of *T. castaneum*. While Hamza et al. [26] found that different concentrations (4, 8, 16, 32, and 64 ppm) of *M. oleifera* leaf ethanolic and aqueous extracts have larvicidal and anti-feeding effects on cotton worm, *Spodoptera littoralis*. With the ethanolic extract, mortality increased between 23.3 and 46.7%, 50.0 and 83.3%, and 56.7 and 96.7%; and the aqueous extract killed 13.3 and 26.7%, 36.7 and 43.3%, and 4.7 and 90.0% in 1, 3, and 5 days, respectively.

In this research work, the number of *T. merganser* hatched eggs decreased according to the concentration increase, indicating that all the concentration rates have residual toxic effect on the eggs laid and/or on *T. merganser* females’ biology, and have a negative effect on the growth rate (r_i_) of the red spider mite. Moawad and Sadek [16] reported that *M. oleifera* oil concentrations of 5, 7.5, and 10% (*v*/*v*) reduce the egg viability of cotton leaf worm, *Spodoptera littoralis*, by 50.07, 75.41, and 88.90, respectively, as compared to the control. Derbalah et al. [27] mentioned that the bioactive compounds with egg-killing effect affect the embryo formation and the developing larvae within the egg’s membrane, before hatching. In this regard, Fetoh and Al-Shammery [28] mentioned that the chemical substances present in plants are able to block out the micropyle region of the egg, impairing gas exchange and leading to the embryo’s death inside the egg. On the other hand, Hosny et al. [29] mentioned that adult mite females exposed to discs treated with acaricides could be subject to partial or temporary sterility, leading to a lower number of eggs laid by females per day, as well as leading to a lower number of viable eggs, as compared to the control. According to the previous information, although more research is required, the extract of *M. oleifera* leaves causes these two effects and could be characterized as having sterilizing effect.

Feed intake was assessed based on the percentage of damage, which decreased as the concentration rates increased. Hamza et al. [26] reported that concentrations of 4.539 and 3.264 ppm of *M. oleifera* aqueous and ethanolic extracts reduced the feed intake of *Spodoptera littoralis* fourth larval instars by 14.43 and 24.51%, respectively. Moawad and Sadek [16] documented that 5% (*v*/*v*) of *M. oleifera* oil inhibited feeding in *Spodoptera littoralis* by 20%, as compared to the control. On the other hand, Tridiptasari et al. [30] found that 20% (*v*/*v*) of *M. oleifera* ethanolic leaf and seed extracts caused an anti-feeding effect on the third (54.66 and 43.62%), fourth (53.57 and 43.33%), and fifth (55.11 and 45.34%) instars of *Spodoptera litura* (Fab.) (Lepidoptera: Noctuidae) at 12 h. Furthermore, Ojo et al. [17] found that *M. oleifera* leaf powder (0.1 at 2 g/20g of cowpea) reduced the damage caused by *Callosobruchus maculatus* to *V. unguiculata* seeds. 

This research work demonstrated that *M. oleifera* has potential to control *T. merganser*, since it lowered its reproduction and survival rates. Rattan [31] mentioned that this successful plant species synthesize a broad range of slightly toxic secondary metabolites, or a small quantity of bioactive highly toxic compounds. Therefore, *M. oleifera* can be considered a successful biopesticide [21]. To conclude, the results of our research work showed that *M. oleifera* has potential as biological control of red spider mites. The extract caused chronic toxicity in females, reducing egg viability, damage by feeding, and the number of eggs laid by *T. merganser* females. Further research is necessary on the behavior and biology of the nymphs (hatched eggs) emerging from the eggs exposed to the residual extract effects; also, to study other developmental stages of this spider mite. Such research might give us a clearer idea of the scope of the effects of *M. oleifera*. Furthermore, other types of *M. oleifera* extracts to control *T. merganser* and other mite species of agricultural importance need further research. It is important to assess the extract against the natural enemies of red spider mites.

## 4. Materials and Methods

### 4.1. Red Spider Mite Colony

A colony of *T. merganser* was obtained from infested papaya plants (*Carica papaya* L. (Caricaceae)), in Ciudad Victoria (23°44’38.4” N and 99°9’57.599” W, 329 m above sea level), Tamaulipas, Mexico. *Tetranychus merganser* was identified according to Baker and Tuttle’s [32] taxonomic key. In order to increase the red spider mite population, *T. merganser* females were placed on bean plants (*Phaseolus vulgaris* L. (Fabaceae)) under greenhouse conditions (29 ± 4 °C and 60 ± 15% relative humidity (RH)).

### 4.2. Preparation of the Extract and the Plant Material

We collected visibly clean leaves of *M. oleifera* in the Applied Ecology Institute at Universidad Autonoma de Tamaulipas, Tamaulipas, Mexico. The Moringa tree was two year old and the leaves were mature. The leaves were dried in a conventional oven (Quincy lab, Chicago, IL, USA model 20GCE-LT) at 60 °C during three days, until obtaining a consistent weight. The sample was ground (Miller CUISINART, Stamford, CT, USA, model DBM-8) until forming particles of 1 mm [33]. The powder was stored in dark bottles at ambient temperature in preparation for the extraction.

During three days, with the aid of a stirrer in total darkness, we blended 14 g of the dry homogenized *M. oleifera* leaf powder with 200 mL of absolute ethanol at ambient temperature. The blend was filtered with Whatman No. 1. filter paper. We used a rotary evaporator (IKA-RV 10 digital V, Staufen Baden Württemberg, Germany) to eliminate the solvent, at temperatures lower than 40 °C, under reduced pressure. Finally, we eliminated the remaining ethanol placing the flask on the kiln drier until obtaining a consistent weight (three days) [34]. The extract was scraped off and stored in Eppendorf tubes and kept in a freezer at −10 °C before conducting the bioassays.

### 4.3. Phytochemical Extract Analysis

We performed the ethanolic extract analysis of *M. oleifera* in order to do the qualitative detection tests of the phytochemicals. The tests included alkaloids (Dragendorff & Sonheschain reagent), phenols (Iron Chloride test FeCl_3_), carbohydrates (Molisch reagent), carotenoids (H_2_SO_4_ and FeCl_3_ reagents), coumarins (Erlich reagent), flavonoids (Shinoda reagent and NaOH at 1%), free reducing sugar (Fehling and Benedict reagent), cyanogenic glycosides (Grignard reagent), purines (HCl test), quinones (NH_4_OH and H_2_SO_4_ reagents by anthraquinone and Börntraguer test by benzoquinone), saponins (foam test, Bouchard reagent for steroid saponins and Rosenthaler reagent), terpenoids (Ac_2_O reagent), soluble starch (KOH and H_2_SO_4_ test), and, finally, tannins (FeCl_3_ reagent and iron cyanide) [35,36,37].

### 4.4. Experimental Design

We used the sand technique described by Ahmadi [38] with a slight modification. Bean discs of 2.5 cm in diameter were cut and placed on water-soaked cotton with the underside facing up. With the help of a sterile punch with a diameter of 2.5 cm, the bean leaf discs were cut. We placed a disc inside a 5 cm-wide Petri dish, with ten *T. merganser* females per bean disc. The experiment was done under laboratory conditions at 27 ± 1 °C and 70–80% relative humidity (RH), with a photoperiod of 12:12 h (light:darkness). The bean leaf discs were split at random in eight groups: a control group and seven treatment groups, one per extract concentration. The bean leaf disc was the replicate. We had three replicates per group. Ten mite females were placed on each bean leaf disc and they were sprayed two times (0.7 ± 0.1 mL per spray) with each concentration, from a distance of 30 cm between the sprayer and the leaf discs. We prepared the different concentrations (0.1, 0.5, 1, 5, 10, 15, and 20 % *v*/*v* (ethanol/water)), starting from an initial solution of 1000 ppm of powdered *M. oleifera* extract in ethanol. Immediately, we used a manual sprayer (Truper^®^ Model 14687, Ciudad de Victoria, Mexico) to apply the different concentrations of the extract. The control treatments were sprayed with water only. We recorded the number of dead mites at 24, 48, and 72 h. Red spider mites with ataxia (active, apparently messy movement) were considered dead, as well as mites lying on their backs, legs up, or without moving. We also counted the number of eggs laid and we observed the damage caused by the spider mite.

On the fourth day, after applying the extract, we recorded the number of hatched eggs per disc, in order to assess the residual effect on the eggs laid during the first 24 h. Ten adult *T. merganser* females and five males were put onto each bean leaf disc to enhance reproduction and oviposition. After 24 h, males were removed, leaving only females. The time that an egg of *T. merganser* takes to hatch is around 3.6 ± 0.3 days, at 27.5 °C on bean discs (*P. vulgaris*), with a photoperiod of 16:8 h light:darkness and 60–70% RH [5]. The hatching time on papaya discs (*C. papaya*) is around 4.10 ± 0.52 days, at 27.0 °C with a photoperiod of 14:10 h L:D and 60 ± 2% RH [39]. The eggs that did not hatch within that time were considered non-viable. We used a dissection microscope (UNICO Stereo and Zoom Microscopes ZM180, Dayton, NJ, USA) to do the counting and damage observations.

### 4.5. Tetranychus merganser Mortality Essay

We corrected the mortality data using Abbott’s formula on the control group [40].
MC = [(%M_T_ − %M_C_)/(100 − %M_C_)] × 100,(1)
where %M_T_ is the mortality percentage in the treated group and %M_C_ is the mortality percentage in the control group.

### 4.6. Tetranychus merganser Oviposition and Hatched Eggs

We used Kramer and Mulla’s formula [41] to determine the oviposition activity percentage (OAP) in each concentration.
OAP = [(N_T_ − N_C_)/(N_T_ + N_C_)] × 100,(2)
where N_C_ is the number of eggs in the control group and N_T_ is the number of eggs in the treated group. The percentage values ranged between +100 and −100. The positive values indicate that we observed more oviposition in the treatment than in the control group (showing that the treatment stimulated oviposition activity). In contrast, more oviposition in the control group than in the treatment resulted in a negative OAP, indicating that the treatment inhibited oviposition.

In order to measure the residual effect of the extract on the spider mites’ egg hatching, we recorded the number of hatched eggs after the fourth day. The hatching percentage was calculated using the next formula (E_24h_/E_4d_) × 100, where E_24h_ is the number of eggs laid on the first 24 h and E_4d_ is the number of eggs that hatched on the fourth day. We used four days as the basis to calculate the time that an egg of *T. merganser* takes to hatch on bean leaf discs [5].

### 4.7. Tetranychus merganser Anti-Feeding

We assessed the anti-feeding effect through the inhibition percentage of *T. merganser* feed intake on bean leaf discs, as compared to the control group. At 72 h, we classified the symptoms observed on the bean leaf discs according to an ordinal scale developed by Hussey and Parr [42] and Nachman and Zemek [43]. We converted them into percentages: 0 = 0% damage (with no feeding damage); 1 = 1–20%; 2 = 21–40%; 3 = 41–60%; 4 = 61–80%; and 5 = 81–100% of feeding damage (dense marks, or wilting, after eating all the bean disc). Kramer and Mulla’s [41] formula criterion was used to measure the females’ feed intake. The positive values show that there was more damage in the treatment group than in the control group, indicating that the treatments stimulated feeding. The negative values represent more severe damage to the control group than to the treatment group, indicating that feeding was inhibited in the treatment group [44].

### 4.8. Tetranychus merganser Growth Population

The growth rate (r_i_) parameter was used to determine the extract’s impact on the red spider mite population. We used the following equation to calculate r_i_:r_i_ = (N_f_/N_O_)/∆t(3)
where N_O_ is the initial number of individuals (10 adult females by replicate), N_f_ is the final number of individuals (adult surviving females plus the eggs laid at the end of the bioassay), and Δt is the number of days elapsed from the beginning until the end of the bioassay (equal to 3 days). Positive values of r_i_ indicate a growing population. The negative values indicate a declining population, and r_i_ = 0 indicates a stable population [45,46]. 

### 4.9. Statistical Analysis

The number of laid eggs, egg viability, the percentage of feeding damage, and the growth rate were statistically analyzed using the analysis of variance (ANOVA) and the means compared with Tukey’s HSD (*p* ≤ 0.05). Mortality percentage was analyzed by using PROC GLM and to separate the means of the treatments we applied Tukey’s HSD (*p* ≤ 0.05) test. Probit analysis was used to estimate the lethal concentration (LC_50(90)_), causing 50(90)% mortality in the *T. merganser* population on the third day after application of the extract, as well as for estimating the confidence intervals of 95% (CI95) for LC_50(90)_ [47]. We used the SAS/STAT program in our analysis [48].

## Figures and Tables

**Table 1 plants-10-01034-t001:** Secondary metabolites (“+” = present; “−” = absent) in *Moringa oleifera* leaf ethanolic extract.

Bioactive Compound		Test	Bioactive Compound		Test
Phenols	+	FeCl_3_	Carbohydrates	+	Molisch’s
Flavonoids	+	Shinoda for flavanone’s	Alkaloids	+	Dragendorff’s
	+	NaOH at 1% for flavanone’s or Xanthone		+	Sonheschain’s
Tannins	+	FeCl_3_ for gallic acid	Quinones	+	H_2_SO_4_ for anthraquinone
	+	Ferrocyanide for phenols		+	Bröntraguer’s for benzoquinone
	+	Jelly		+	NH_4_OH for anthraquinone
Saponins	+	Bouchard for steroidal saponins	Carotenoids	+	H_2_SO_4_ and FeCl_3_ reagents
	−	Foam	Sugar reducers	+	Fehling’s
	−	Rosenthale		+	Benedict’s
Coumarins	+	Erlich’s	Cyanogenic glycosides	+	Grignard’s
Purines	−	HCl	Terpenoids	−	Ac_2_O
Soluble starch	+	KOH and H_2_SO_4_			

**Table 2 plants-10-01034-t002:** Effect of *Moringa oleifera* leaf ethanolic extract at different concentrations on *Tetranychus merganser* females under controlled laboratory conditions (*n* = 3 with 10 red spider mites per replicate).

Treatments	Corrected Mortality (±SE) Average *		
	24	48	72
0.1	0.00 ± 0.00 e	6.67 ± 3.33 e	13.70 ± 3.16 c
0.5	10.00 ± 0.00 d	10.00 ± 0.00 e	20.74 ± 0.74 c
1	13.33 ± 3.33 cd	20.00 ± 0.00 d	27.41 ± 2.59 c
5	23.33 ± 3.33 bc	33.33 ± 3.33 cd	55.19 ± 2.89 b
10	30.00 ± 0.00 b	43.33 ± 3.33 bc	68.89 ± 5.88 b
15	43.33 ± 3.33 a	56.67 ± 6.67 b	92.96 ± 3.53 a
20	46.67 ± 3.33 a	86.67 ± 3.33 a	96.30 ± 3.70 a

* Mean values and ± standard errors (SE) are presented. Different letters within a column indicate significant differences (*p* < 0.05; ANOVA and Tukey’s HSD test).

**Table 3 plants-10-01034-t003:** Estimation of the LC_50(90)_ for ethanolic extract of *Moringa oleifera* leaves on *Tetranychus merganer* females, after the third day, with *n* = 3 and 10 red spider mites per replicate.

LC_50_(CI_95_)	LC_90_(CI_95_)	b ± SE	χ^2^	Pr > χ^2^
6.06	17.59	4.75 ± 0.86	30.52	<0.0001
(4.25–7.39)	(14.51–24.74)			

LC: Lethal concentration that kills 50% (90%) of the population. CI: Confidence interval at 95%. b: slope ± standard error. χ^2^: Chi-square value.

**Table 4 plants-10-01034-t004:** Effects of *Moringa oleifera* leaf ethanolic extract on oviposition and the growth rate of *Tetranychus merganser*.

Concentrations (% *v*/*v*)	Eggs Number ± SE *	OAP (%) ± SE	Eggs Number ± SE	OAP(%) ± SE	Eggs Number ± SE	OAP (%) ± SE	Growth Rate ± SE
	24 h		48 h		72 h		
Control	61.33 ± 0.88 a		142.00 ± 2.08 a		222.33 ± 1.76 a		1.04 ± 0.00 a
0.1	58.33 ± 1.76 a	−2.54 ± 1.31	100.00 ± 1.15 b	−17.35 ± 0.51	138.67 ± 1.86 b	−23.18 ± 0.92	0.89 ± 0.00 b
0.5	47.00 ± 0.58 b	−13.23 ± 0.56	81.67 ± 0.88 c	−26.96 ± 1.16	107.33 ± 1.76 c	−34.89 ± 0.95	0.81 ± 0.00 b
1	30.00 ± 0.58 c	−34.31 ± 0.66	46.67 ± 0.88 d	−50.52 ± 1.07	58.67 ± 0.88 d	−58.25 ± 0.33	0.62 ± 0.00 c
5	28.00 ± 0.58 c	−37.30 ± 1.50	32.67 ± 1.45 e	−62.61 ± 1.36	33.33 ± 0.88 e	−73.93 ± 0.53	0.44 ± 0.01 d
10	4.33 ± 0.33 d	−86.83 ± 0.77	13.33 ± 0.33 f	−82.82 ± 0.62	14.00 ± 0.58 f	−88.15 ± 0.53	0.18 ± 0.01 e
15	2.33 ± 0.33 d	−92.66 ± 1.09	4.00 ± 0.58 g	−94.54 ± 0.72	4.67 ± 0.67 g	−95.89 ± 0.57	−0.22 ± 0.05 f
20	1.67 ± 0.33 d	−94.69 ± 1.09	2.67 ± 0.33 g	−96.31 ± 0.47	2.67 ± 0.33 g	−97.63 ± 0.30	−0.40 ± 0.00 g

* Mean values and ± standard errors (SE) are presented. Different letters indicate significant differences (*p* < 0.05; ANOVA and Tukey’s HSD test). OAP, percentage of oviposition activity.

**Table 5 plants-10-01034-t005:** Mean values (±SE) of hatched eggs laid by *Tetranychus merganser* females during the first 24 h.

Concentration % (*v*/*v*)	Mean Values of Hatched Eggs *	Mean Values Hatching Percentage
Control	59.33 ± 0.88 a	96.74 ± 0.05
0.1	46.33 ± 1.20 b	79.45 ± 0.39
0.5	32.67 ± 0.67 c	69.49 ± 0.79
1	19.00 ± 0.58 d	63.31 ± 0.71
5	14.00 ± 0.58 e	49.96 ± 1.03
10	1.00 ± 0.00 f	23.33 ± 1.67
15	0.00 ± 0.00 f	0.00 ± 0.00
20	0.00 ± 0.00 f	0.00 ± 0.00

* Mean values and ± standard errors (SE) are presented. Different letters indicate significant differences (*p* < 0.05; ANOVA and Tukey’s HSD test).

**Table 6 plants-10-01034-t006:** Effects of *Moringa oleifera* leaf ethanolic extract of on *Tetranychus merganser* feeding rate.

Concentration % (*v*/*v*)	Average Damage (%) ± SE	Food Intake (%) ± SE	Average Damage (%) ± SE	Food Intake (%) ± SE	Average Damage (%) ± SE	Food Intake (%) ± SE
	24 h *		48 h		72 h	
Control	15.67 ± 1.20 a		20.00 ± 0.58 a		34.00 ± 0.58 a	
0.1	12.00 ± 0.58 b	−13.05 ± 3.75	20.67 ± 0.33 b	−13.27 ± 1.78	29.00 ± 0.58 b	−7.94 ± 1.59
0.5	9.33 ± 0.67 bc	−25.04 ± 6.78	17.00 ± 0.58 c	−22.74 ± 2.30	20.00 ± 0.58 c	−25.93 ± 1.88
1	7.33 ± 0.88 cd	−36.57 ± 1.93	10.00 ± 0.58 d	−45.97 ± 2.82	17.33 ± 0.88 c	−35.51 ± 2.82
5	4.67 ± 0.33 de	−53.67 ± 5.04	6.67 ± 0.67 e	−60.56 ± 2.53	10.667 ± 1.20 d	−52.42 ± 4.19
10	2.33 ± 0.33 e	−73.73 ± 4.62	3.33 ± 0.33 f	−78.10 ± 1.56	5.00 ± 0.58 e	−74.39 ± 2.82
15	2.00 ± 0.58 e	−77.05 ± 6.66	2.67 ± 0.33 f	−82.05 ± 2.11	3.67 ± 0.33 e	−80.55 ± 1.65
20	1.67 ± 0.33 e	−80.31 ± 4.60	2.33 ± 0.33 f	−84.19 ± 1.78	2.67 ± 0.33 e	−85.48 ± 1.72

* Mean values and ± standard errors (SE) are presented. Different letters indicate significant differences (*p* < 0.05; ANOVA and Tukey’s HSD test).

## Data Availability

The data presented in this study are available on request from the corresponding author.

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
