# Peer review of "Bioacaricidal Potential of Moringa oleifera Ethanol Extract for Tetranychus merganser Boudreaux (Acari: Tetranychidae) Control"

_plants, 2021, doi:10.3390/plants10061034_

Round 1

Reviewer 1 Report

Heinz-Castro et al. Bioacaricidal Potential of Moringa oleifera Ethanol Extract for 2 Tetranychus merganser Boudreaux (Acari: Tetranychidae) Control

This paper describes the effects of an ethanol extract of leaves of Moringa oleifera on the red spider mite.  The results suggest a clear dose-response in terms of the concentration of extract used and the adverse effects observed on the mite. Thus, overall, the paper provides a good example of a botanical extract that may have use in pest control.

On the negative side, some of the writing is bit awkward and some important details are missing (see below).  Importantly, all the results seem to be based on a single laboratory experiment, with only three replicates per concentration, and, it seems, only one batch of the leaf extract. There has been no attempt to assess whether there might be variation between batches of extract, or whether the effects on the mites would be seen if the plant extract was applied to whole potted plants or under glasshouse or field conditions.  Additionally there is no indication of the duration of any negative effects.  I am also slightly concerned about the statistics, and the seeming lack of variation in any of the response variables as indicated by the exceedingly small standard errors.

Specific Comments

Abstract

Ln 27   is ‘human friendlier’ really needed here?

Ln 30  I don’t think the 0.1v/v data are needed here – just present the 20% data for simplicity.

Ln 32  rather than ‘depended on’ could you just say ‘were all reduced at high concentrations’

Introduction

Ln 38  you have left some of the template text in the paper

Ln 58 interesting that such an edible plant has such insecticidal properties?

Ln 60 this sentence needs sorting

Methods

Ln 206 italics for mite name

Ln 210 location where the leaves collected?  Old leaves or young leaves?

Ln 222 how did you pour the extract if it was completely dry?

Ln 225 italics missing

Ln 237  2.5 mm or 2.5 cm ?  How did you cut the discs?

Ln 238 italics missing

Ln 241 no positive control?

Ln 246 it is not clear how you have made up these extracts – 20% means 20% of the ethanol extract that was centrifuged to dryness, in 80% of water?  Have you missed out a step somewhere?

Ln 250 so the eggs are the total on the disk, regardless of how many mites were alive or dead?  This is Ok need to clarify it is not eggs per mite.

Ln 252 again hatched eggs per disk.  What if there no eggs?

Ln 301 mortality is binomial so, strictly, a Generalized Linear Model should be used.

Ln 304 what day was LC50 calculated?

Results

Table 2  Title should explain these results apply to leaf disk assay under controlled laboratory conditions, and the sample size = 3, with 10 mites per replicate.  The SEs are very small for 3 replicates.  Suggests mortality for the replicates in each treatment was almost identical.

Table 3. Title needs to better explain what the table is shown, and how many mites were exposed at each dose.  in the foot note you have CE but the table shows CL ?   At what time point were the LC50 and LC90 calculated?

Ln 102 the females can’t lay eggs if they are already dead.  The eggs were produced by the cohort of females (while alive).

Ln 106 growth rate of the spider mite population

Table 4.  Again, the standard errors seem very small for three replicates when you are talking about variation in egg laying.

Discussion

Ln 140 paragraph .  Put in full genus names of all the other insects tested.

Author Response

Dear Reviewer 1

The following table shows the suggestions made to the manuscript by the reviewers, as well as the answers to them. It should be noted that there was a general review throughout the writing in the use of commas, prepositions, passive voice, and some connecting words ideas. For correction were used the MS Word Track Changes function, as indicated.

The study was carried out in a single experiment because it is required to study the effect of the botanical extract on some biological aspects of the mite, such as mortality, oviposition, feeding (damage) and population growth, as well as whether the laid eggs can be affected by the residuality of the extract. These experiments can also be done separately but in the laboratory, conditions, but the reality, what is intended is to know, apart from killing the mite, what else, is the contribution of the extract on a group of mites. Which leads to an approximation of what would happen in field or greenhouse conditions.

With regard to the small variation (SEs), this yielded the data, it may be the homogeneous conditions in which the experiment was carried out.

Reviewer observation

Reply

Text correction

Abstract

Ln 27   is ‘human friendlier’ really needed here?

not necessary in the abstract.

So it was removed.

There is a need for alternate control measures that are environmentally friendlier than chemical pesticides

Ln 30  I don’t think the 0.1v/v data are needed here – just present the 20% data for simplicity.

We agree with the reviewer because a higher percentage of mortality was observed at 20% of the extract concentration.

So it was removed.

Mites treated with 20% (v/v) killed 86.67%, 13.70% and 96.30%, at 24, 48 and 72 h, respectively, as compared to the control treatment.

Ln 32  rather than ‘depended on’ could you just say ‘were all reduced at high concentrations’

This suggestion is another way to describe the results, but we agree with the suggestion.

So it was romoved

Oviposition, egg hatching and the damage caused by red spider mites were all reduced at high concentrations

Introduction

Ln 38  you have left some of the template text in the paper

The part of the template text has already been removed

Ln 58 interesting that such an edible plant has such insecticidal properties?

If it is interesting but the consumption is not in large quantities so it is not harmful to humans.

Ln 60 this sentence needs sorting

The sentence wording was improved

The original research works that studied the different parts of M. oleifera tree, show different medicinal properties, such as anti-tumor, anti-pyretic, anti-epileptic, anti-inflammatory, anti-ulcer, anti-diabetic, anti-oxidizing, anti-fungal, anti-bacterial and anti-carcinogenic effects

Methods

Ln 206 italics for mite name

The scientific name of the red spider mite was put in italics

T. merganser

Ln 210 location where the leaves collected?  Old leaves or young leaves?

They were collected at the Institute of Applied Ecology at the Autonomous University of Tamaulipas. And the moringa tree was two years old and the leaves were mature.

We collected Visibly visibly clean leaves of M. oleifera were collected in the Applied Ecology Institute at Universidad Autonoma de Tamaulipas, Tamaulipas, Mexico. The Moringa tree was two year old and the leaves were mature.

Ln 222 how did you pour the extract if it was completely dry?

Translation error

The extract was scraped off and stored into Eppendorf tubes and kept in a freezer at −10 °C, before conducting the bioassays.

Ln 225 italics missing

The scientific name of the plant was put italicized. Corrected.

M. oleifera

Ln 237  2.5 mm or 2.5 cm?  How did you cut the discs?

Is 2.5 cm.

With a sterile punch.

The correct is 2.5 cm

With the help of a sterile punch with a diameter of 2.5 cm, the bean leaf discs were cut.

Ln 238 italics missing

The scientific name of the red spider mite was put in italics

T. merganser

Ln 241 no positive control?

We did not use a positive control (acaricide or insecticide) because the objective was to evaluate the effect of the ethanolic extract of M. oleifera leaf, and not to compare whether the extract is more efficient or effective on red spider mites than the chemical control.

Ln 246 it is not clear how you have made up these extracts – 20% means 20% of the ethanol extract that was centrifuged to dryness, in 80% of water?  Have you missed out a step somewhere?

The procedure for formulating concentrations is clarified.

A manual sprayer (Truper® Model 14687 Ciudad de Victoria, Mexico) was used to apply M. oleifera extract at different concentrations. An initial solution of 1000 ppm of powdered was prepared in ethanol from which the following dilutions prepared concentrations, 0.1, 0.5, 1, 5, 10, 15, 20 % v/v (ethanol/water)

Ln 250 so the eggs are the total on the disk, regardless of how many mites were alive or dead?  This is Ok need to clarify it is not eggs per mite.

Ln 252 again hatched eggs per disk.  What if there no eggs?

The wording was improved.

Ten adult T. merganser females and five males were put onto each bean leaf disc to enhance reproduction and oviposition.  After 24 hours, males were removed, leaving only females

Ln 301 mortality is binomial so, strictly, a Generalized Linear Model should be used.

From the beginning, we used PROC GLM to analyze mortality data, but we did not specify that we use this procedure.

Mortality percentage was analyzed by using PROC GLM and to separate the means of the treatments we applied Tukey’s HSD (P ≤ 0.05) test

Ln 304 what day was LC50 calculated?

It was calculated on the third day.

Probit analysis was used to estimate the lethal concentration (CL50(90)), causing 50(90) % of mortality in T. merganser population on the third day after application of the extract as well as confidence intervals of 95% (IC95) for CL50(90) [47].

Results

Table 2  Title should explain these results apply to leaf disk assay under controlled laboratory conditions, and the sample size = 3, with 10 mites per replicate.  The SEs are very small for 3 replicates.  Suggests mortality for the replicates in each treatment was almost identical.

The title of Table 2 is modified as indicated by the reviewer.

The SEs were almost identical in each treatment because the study was carried out under controlled laboratory conditions and the extract was applied homogeneously.

Effect of Moringa oleifera leaf ethanolic at different concentrations on Tetranychus merganser females under controlled laboratory conditions and n=3 with 10 red spider mites per replicate.

Table 3. Title needs to better explain what the table is shown, and how many mites were exposed at each dose.  in the foot note you have CE but the table shows CL ?   At what time point were the LC50 and LC90 calculated?

The title of Table 3 is modified as indicated by the reviewer.

Estimation of the LC50(90) for ethanolic extract of Moringa oleifera leaves on Tetranychus merganer females, after third day with n=3 and 10 red spider mites per replicate.

Ln 102 the females can’t lay eggs if they are already dead.  The eggs were produced by the cohort of females (while alive).

It's true.

The eggs were laid by females who could still oviposition.

We rewrite sentence to clarify the idea.

Egg production by T. merganser females that could still oviposit was significantly different at 24, 48 and 72 h

Ln 106 growth rate of the spider mite population

Table 4.  Again, the standard errors seem very small for three replicates when you are talking about variation in egg laying.

The SEs were almost identical in each treatment because the study was carried out under controlled laboratory conditions and the extract was applied homogeneously.

Discussion

Ln 140 paragraph .  Put in full genus names of all the other insects tested.

The full names of the insect species (genus and species) were written

Reviewer 2 Report

This manuscript represents an evaluation of the effects of plant extracts on spider mites. It is clear and well written. The introduction should consider more in depth progresses obtained with plant extracts in controlling pests. Material and method and results sections are clearly described. Perhaps the discussion should consider more in details issues with acaricide resistance and the impact of these new compounds in IPM. Minor remarks are reported in an annotated version of the ms.

Author Response

Dear Reviewer 2

The following table shows the suggestions made to the manuscript by the reviewers, as well as the answers to them. It should be noted that there was a general review throughout the writing in the use of commas, prepositions, passive voice, and some connecting words ideas. For correction were used the MS Word Track Changes function, as indicated.

Reviewer commented,

This manuscript represents an evaluation of the effects of plant extracts on spider mites. It is clear and well written. The introduction should consider more in depth progresses obtained with plant extracts in controlling pests. Material and method and results sections are clearly described. Perhaps the discussion should consider more in details issues with acaricide resistance and the impact of these new compounds in IPM. Minor remarks are reported in an annotated version of the ms.

Regarding the annotations within the manuscript, dear reviewer we could not see them since the document was not found in the system. But we made improvements to the manuscript. See the paper.

Reviewer 3 Report

Botanicals are becoming more important in an integrated pest management with alternative control mechanisms. The present paper describes the use of an ethanolic extract of Moringa oleifera leaves against red spider mites, Tetranychus merganser. 

Results show, that the ethanolic extract inhibits oviposition and egg hatching of T. merganser in a dose-dependent manner, but also the feeding rate of adult females.

Such kind of results are not surprising and have been shown for hundreds of plant extracts on hundreds of pest insects.  The results would gain a lot in importance if the authors had not only made a rough characterization of the leaf compounds, but a thorough analysis, e.g. using mass spectrometry. Some studies on secondary metabolites of M. oleifera had already been done (see Ref. 14) and should be discussed in more detail here.

Minors: 

  • the first sentence of the Introduction is trivial and should be deleted
  • legend to Table 2: do authors mean "different letters within a column"?
  • the first sentence of the Discussion is also trivial
  • line 206 and others: give all species names in italics, also in the References
  • line 230: write chemical formulas correctly
  • line 231; use lowercase letters for names of chemical compounds, unless you mean trade names
  • line 240: 12:12 h or hrs
  • line 257: what do authors mean with L:O?
  • line 261: T. merganser

Author Response

Dear Reviewer 3

The following table shows the suggestions made to the manuscript by the reviewers, as well as the answers to them. It should be noted that there was a general review throughout the writing in the use of commas, prepositions, passive voice, and some connecting words ideas. For correction were used the MS Word Track Changes function, as indicated.

Regarding reference 14, it was included in the Introduction as well as the Discussion. In the Discussion, it is mentioned that the findings found in reference 14 are similar to those found in our study. It should be mentioned that the application made in reference 14 is on Hela cells. But with regard to secondary metabolites, it was cited. And the wording was improved.

Reviewer observation

Reply

Text correction

·         the first sentence of the Introduction is trivial and should be deleted

legend to Table 2: do authors mean "different letters within a column"?

Its true

Different letters within a column indicate significant differences (P < 0.05; ANOVA and Tukey’s HSD test).

·         the first sentence of the Discussion is also trivial

We do not consider it trivial. It is important to mention that other studies found the same secondary metabolites.

line 206 and others: give all species names in italics, also in the References

All scientific names were italicized throughout the manuscript.

line 230: write chemical formulas correctly

Corrected formula

NH4OH

line 231; use lowercase letters for names of chemical compounds, unless you mean trade names

name of chemical compound corrected

anthraquinone

line 240: 12:12 h or hrs

It´s “h”

12:12 h

line 257: what do authors mean with L:O?

It´s “L:D”

14:10 h L:D

line 261: T. merganser

The spider mite species name has been corrected.

Tetranychus merganser

Round 2

Reviewer 1 Report

Many of the corrections suggested by the reviewers have been addressed in an acceptable manner.  However, there are still some minor English errors that need correcting.  

Author Response

Dear Reviewer 1

Reviewer observation

However, there are still some minor English errors that need correcting. 

Raply

It should be noted that there was a general review throughout the writing in the use of commas, prepositions, passive voice, and some connecting word ideas. For correction were used the MS Word Track Changes function, as indicated.

Reviewer 3 Report

Authors have improved the manuscript, but I still think that the results would gain a lot in importance if the authors had not only made a rough characterization of the leaf compounds, but a thorough analysis. 

Minor corrections:

  • Table 3: use LC (and not CL) in the entire table
  • Tabele 5: delete the second "laid" in the table legend; delete the last column  (non-hatching percentage), which is just the difference to 100 % from the third column); say "mean values" in all figure and table legends
  • line 267: of what?
  • line 268: sentence is not complete

Author Response

Dear Reviewer 3

The following table shows the suggestions made to the manuscript by the reviewers, as well as the answers to them. It should be noted that there was a general review throughout the writing in the use of commas, prepositions, passive voice, and some connecting word ideas. For correction were used the MS Word Track Changes function, as indicated.

Reviewer observation

Authors have improved the manuscript, but I still think that the results would gain a lot in importance if the authors had not only made a rough characterization of the leaf compounds, but a thorough analysis.

Reply

We agree with the reviewer's observation. Although the results of the secondary metabolites are an approximation, these showed the presence of important phytochemicals in the control of arthropods. Unfortunately, we do not have the equipment to carry out said analysis (spectrometer), also at that time, we did not have the financial resources to pay for the services of a laboratory to perform the analyzes. At this time we only have the resources to pay the bill for the publication cost if the article were accepted for publication in Plants journal.

Table 3: use LC (and not CL) in the entire table

Reply

We corrected it in table 3 and in the methodology section

Tabele 5: delete the second "laid" in the table legend; delete the last column  (non-hatching percentage), which is just the difference to 100 % from the third column); say "mean values" in all figure and table legends

Reply

"laid" was removed

ast column  (non-hatching percentage) was removed

We included "mean values" in all figure and table 5 legends

line 267: of what?

line 268: sentence is not complete

Reply

We improve the writing